# ON 10X BETTER SCALABILITY:
# KV STORES SCALE UP KV CACHE

## ABSTRACT

Large language models (LLMs) rely on Key-Value (KV) cache to reduce time-to-first-token (TTFT) latency, but existing disk-based KV cache systems using file-per-object layouts suffer from severe scalability bottlenecks due to file system metadata overhead, I/O inefficiency, and poor spatial locality. This paper presents SGLANG-LSM, a database-inspired system that leverages Log-Structured Merge-tree (LSM-tree) architectures for scalable KV cache management. SGLANG-LSM implements a layered system design with three coordinated components: (1) a prefix-preserving storage engine that maintains token sequence locality while efficiently storing large KV cache tensors through key-value separation, (2) an adaptive controller that dynamically optimizes LSM-tree configurations based on shifting workload characteristics, and (3) runtime services including batch operations and automatic resource management for production deployment. Evaluation on large-scale dynamic workloads demonstrates that SGLANG-LSM significantly improves cache hits by up to 143% and reduces TTFT by up to 24% compared to state-of-the-art systems, representing the first systematic application of database storage architectures to large-scale LLM cache management.

## 1 INTRODUCTION

Large language models (LLMs)Achiam et al. (2023); Touvron et al. (2023) has transformed daily life and production, enabling applications such as programming assistants Chen et al. (2021) and universal chatbots OpenAI.. However, widespread adoption of powerful models has introduced operational challenges. Deploying and operating LLM services is increasingly costly due to the high volume of user requests and limited hardware resources Li et al. (2023); Wang et al. (2025b). Exacerbating the pressures, LLM applications typically prepend extensive context-rich prefixes to queries before inference to improve response quality Gao et al. (2024). These prefixes may include retrieved documents Lewis et al. (2020); Gao et al. (2023), conversation history Gao et al. (2024), or few-shot examples Lu et al. (2023) to guide output format. While context-enriched requests enhance accuracy, they substantially increase response-generation latency, particularly the time to first token (TTFT). This latency penalty arises because transformer computation grows superlinearly with input length Vaswani et al. (2017), making longer prefixes disproportionately expensive to process.

A common optimization is to cache and reuse Key-Value (KV) states of repeated prefixes, thereby avoiding redundant computation to reduce TTFT Jin et al. (2024); Gim et al. (2024). Among these approaches, SGLang Zheng et al. (2024) has emerged as a state-of-the-art LLM serving system that achieves superior performance in inference throughput and latency. SGLang employs RadixAttention, a sophisticated KV cache management mechanism that organizes cached states in a hierarchical radix tree structure, enabling efficient prefix sharing and automatic cache reuse across complex sharing patterns. This mature system has become widely adopted in production environments due to its robust performance and scalable architecture. However, conventional approaches that retain KV caches in GPU or CPU memory face severe scalability challenges. As the number of requests and sequence lengths grow, memory pressure inevitably forces eviction Gao et al. (2024); Chen et al. (2025), lowering cache hit rates and diminishing benefits.

To overcome memory limits, recent systems Zheng et al. (2024); Qin et al. (2024); Yao et al. (2024) have begun offloading KV caches to local disk. Further optimizations explore communication scheduling Gao et al. (2024) or selective loading based on cache importance Chen et al. (2025). Yet, building an efficient disk-based KV cache store introduces new challenges. Current designs typically

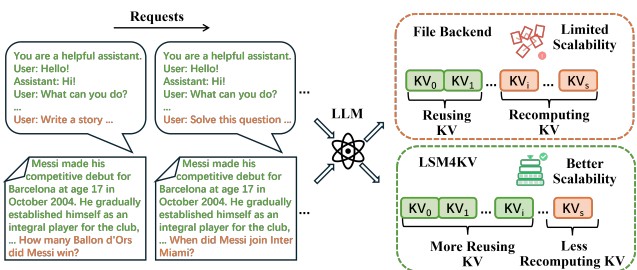

Figure 1: SGLang-LSM.

hash each token in a sequence and create a separate file for the KV-cache tensor generated for that token Zheng et al. (2024); Qin et al. (2024); Yao et al. (2024); Kwon et al. (2023). This file-per-object layout quickly becomes problematic: (1) file system scalability—directories with millions of tiny files suffer from severe metadata overhead; (2) I/O inefficiency—every access incurs open/read/close system calls with little opportunity for batching; (3) lack of spatial locality—related KV states are scattered, preventing sequential I/O and underutilizing disk bandwidth.

These problems strongly resemble classic issues in key–value [1] storage, a domain extensively studied in the database community Idreos & Callaghan (2020). Database systems have long addressed these challenges with mature data structures and workload-adaptive mechanisms. Indeed, database techniques have already been widely adopted in LLM systems, such as vector databases for retrieval Wang et al. (2021); Jayaram Subramanya et al. (2019). However, despite KV caching being naturally a key–value workload, mature storage structures from the database literature have rarely been applied. One reason is historical: current LLM serving systems often operate below the scale where local file system bottlenecks dominate. But as cache repositories expand to hundreds of millions of tokens on local disk, such bottlenecks become unavoidable, making a database-inspired rethinking necessary.

Among the most widely adopted data structures for key–value storage is the Log-Structured Merge-tree (LSM-tree) O'Neil et al. (1996); Dayan & Idreos (2018); Dayan et al. (2017); Luo & Carey (2020). LSM-trees maintain data in a hierarchical structure where most entries reside on disk, making them memory-efficient while supporting high-throughput writes. Through background compaction, they merge and reorganize data to preserve efficient lookups despite continuous updates. These properties make LSM-trees particularly suitable for KV cache management in LLM serving, where workloads involve frequent inserts and lookups. Moreover, cache reuse patterns vary over time Wang et al. (2025a); sharegpt., shifting the read/write balance; while prior studies have explored dynamic tuning of LSM-trees to handle such workload changes Mo et al. (2023); Yu et al. (2024a), our work adapts these insights to the emerging context of large-scale LLM cache storage.

In this paper, we propose SGLang-LSM, a *database-inspired system* that can work as a drop-in storage plugin for SGLang, replacing its disk backend with LSM-based storage. SGLang-LSM addresses the scalability bottlenecks of existing file-per-object approaches through a layered architecture that preserves prefix semantics while leveraging proven database storage techniques. The system consists of three coordinated components: a prefix-preserving storage engine that employs key-value separation to handle large KV tensors efficiently, an adaptive controller that continuously optimizes system performance under dynamic workload conditions, and runtime services that provide operational optimizations for production deployment. The contributions can be summarized as:

- We design and implement SGLang-LSM, a system that applies database storage architectures to large-scale KV cache management, demonstrating how proven storage techniques can address LLM serving challenges.
- We develop a prefix-preserving storage engine with key-value separation that maintains semantic structure while eliminating file system scalability bottlenecks.
- We implement an adaptive controller that dynamically optimizes LSM-tree configurations based on workload characteristics, and runtime services that integrate seamlessly with existing LLM serving infrastructure.
- We demonstrate significant performance improvements on large-scale dynamic workloads, achieving up to 143% higher cache hit rates and 24% lower TTFT compared to state-of-the-art approaches.

---

[1] For clarity, We use *KV* for KV cache terms and *key-value* for storage terms.

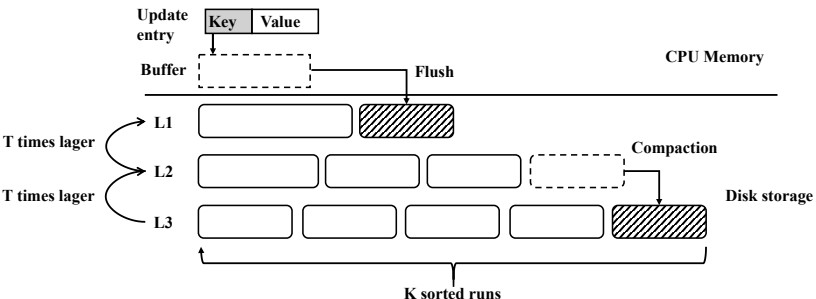

Figure 2: An overview of an LSM-tree.

## 2 PRELIMINARY

### 2.1 GENERATIVE LLM INFERENCE

**Transformer Architecture.** Modern generative LLMs such as GPTs Achiam et al. (2023); Floridi & Chiriatti (2020) and LLaMAs Touvron et al. (2023) are built upon the autoregressive transformer architecture Vaswani et al. (2017); Han et al. (2021). During inference, these models process user prompts as sequences of tokens and generate responses by predicting subsequent tokens using the context of all prior tokens. The transformer consists of $\ell$ layers, each containing self-attention and feed-forward network (FFN) components.

For input token embeddings $X = [x_1, x_2, \ldots, x_s]$, where each $x_i$ represents the embedding vector of the $i$-th token, $s$ is the sequence length, and $d_{model}$ is the model dimension, each layer applies projections using learned weight matrices $W^Q$, $W^K$, and $W^V$ to generate queries, keys, and values: $Q = XW^Q$, $K = XW^K$, $V = XW^V$. The attention computation is: $\text{Attention}(Q, K, V) = \text{softmax}\left(\frac{QK^T}{\sqrt{d_k}}\right) V$, where $\sqrt{d_k}$ is used for scaling to prevent extreme gradients.

**KV Cache Reuse.** Within the attention computation process, each token at position $i$ produces intermediate key and value tensors $K_i$ and $V_i$ (or multi-dimensional tensors for multi-head attention). These tensors represent the contribution of token $i$ to the attention mechanism and are essential for computing attention scores with all subsequent tokens. During autoregressive generation, when generating the $(t + 1)$-th token, the model requires access to the key and value tensors of all preceding tokens $\{K_1, V_1\}, \{K_2, V_2\}, \ldots, \{K_t, V_t\}$ to compute attention scores correctly. Rather than recomputing these tensors from scratch for each new token, modern LLM inference systems cache these intermediate states in GPU memory, forming what is known as the KV cache.

A key optimization opportunity arises when multiple requests share common prefixes. Since computation of KV tensors depends only on the sequence of preceding tokens, KV states computed for a prefix can be directly reused for any request that extends that prefix. For example, if request A processes tokens $[t_1, t_2]$ and request B processes $[t_1, t_2, t_3]$, then the KV cache computed for the prefix $[t_1, t_2]$ can be reused, and only the additional token $[t_3]$ need new computation.

One typical paradigm for managing KV cache reuse employs a hierarchical radix tree structure Zheng et al. (2024); Kwon et al. (2023); Chen et al. (2025); Gao et al. (2024), where each tree node stores a token sequence prefix and each edge stores the associated KV cache tensors for the tokens along that path. When processing a new request, the system performs longest-prefix matching against the tree to identify the maximum reusable KV cache, avoiding redundant computation for shared prefixes. New tokens extend existing branches or create new ones, while an LRU eviction policy removes least-recently-used branches when memory pressure occurs. This hierarchical structure can span multiple storage tiers—from GPU memory for hot cache to CPU memory and even disk storage for cold cache, enabling automatic reuse across complex sharing patterns.

### 2.2 LOG-STRUCTURED MERGE TREES

**LSM-Tree Structure.** Log-Structured Merge-trees (LSM-trees) are a widely adopted data structure for key-value storage that optimize for write-heavy workloads O'Neil et al. (1996). Its design philosophy makes them suitable for KV cache management in LLM serving, where workloads involve frequent cache insertions and lookups. LSM-trees achieve high write throughput by initially buffering all updates in memory and periodically flushing them to disk as sorted runs, avoiding the random I/O patterns that plague traditional storage structures Dayan & Idreos (2018); Dayan et al.

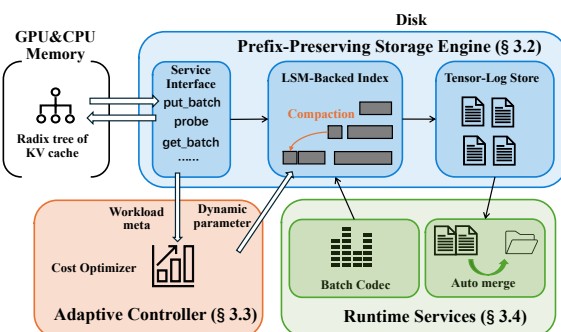

Figure 3: SGLANG-LSM system.

(2017). As shown in Figure 2, an LSM-tree organizes data using an in-memory write buffer and multiple on-disk levels with capacities increasing exponentially by a size ratio $T$. To store $N$ entries of size $e$, an LSM-tree incorporates $L = \log_T \frac{N \cdot e}{M}$ levels, where $M$ is the write buffer size. This hierarchical organization ensures that most data resides on disk while maintaining memory efficiency.

A key mechanism in LSM-trees is compaction, where data from smaller levels is merged and moved to larger levels to maintain the tree structure and remove obsolete entries. This process ensures data eventually propagates through all levels while maintaining sorted order within each level. LSM-trees can employ different compaction policies controlled by the parameter $K$, which represents the maximum number of sorted runs allowed at each level. When $K = 1$, the system uses leveling policy, while $K = T - 1$ corresponds to tiering policy, allowing LSM-trees to balance between write amplification and read performance.

The fundamental advantage of LSM-trees lies in their ability to convert random writes into sequential writes, dramatically improving write throughput on modern storage devices. By deferring the cost of maintaining sorted order through background compaction operations, LSM-trees can sustain high write rates while still supporting efficient lookups through Bloom filters and fence pointers.

**Updates.** LSM-tree uses an out-of-place update strategy where key-value pairs are initially stored in the write buffer. When this buffer is full, entries are flushed to disk. In the worst case, an entry reaches the largest level after $T \cdot L$ compaction processes, resulting in an update cost of $O\left(\frac{TL}{BK}\right)$.

**Point Lookups.** A point lookup in an LSM-tree searches for an entry using a given key. It checks through all levels, returning the result once the matched entry is found. To speed up this process, Bloom filters Bloom (1970) are employed to indicate the presence of the matched entry with false positive rate $p$. Thus the lookup cost is $O(KLp)$ if the entry is absent, or $O(KLp + 1)$ if present.

**Range Lookups.** A range lookup retrieves entries within a key range in an LSM-tree. It incurs $O(KL)$ I/O cost to seek qualified entries across all levels Dayan & Idreos (2018); Dayan et al. (2017), and requires an additional $O(\frac{d}{B})$ cost for entry retrieval, where $d$ is the number of matched entries.

## 3 SYSTEM DESIGN OF SGLANG-LSM

### 3.1 SYSTEM ARCHITECTURE

SGLANG-LSM is designed as a system to support disk-resident KV cache management at serving scale, where the working set routinely exceeds GPU and host memory. The architecture (Figure 3) follows a layered design that exposes a simple, prefix-aware interface to inference schedulers while internally coordinating storage, control, and runtime services. At its core, the system retains the semantic structure of token prefixes rather than flattening them into unordered keys, ensuring that storage operations directly align with the access patterns of LLM inference.

At the foundation lies the Prefix-Preserving Storage Engine, which organizes all persistent KV-cache states. This engine is itself decomposed into three cooperating layers. The service interface layer provides the public contract of the system and records workload metadata that can be fed into higher-level tuning. The disk index layer implements durable ordering of keys using LSM-tree structure, encoding token prefixes so that lexicographic order coincides with prefix order. It stores only compact metadata rather than tensors, enabling efficient compaction. Finally, the tensor-log layer handles

large immutable KV tensors, appending them to sequential data files and supporting scatter–gather reads. The decoupling between index and data ensures that compaction never rewrites large tensor payloads, bounding write amplification and stabilizing I/O performance.

Above the storage engine, the Adaptive Controller continuously monitors workload characteristics and tunes structural parameters of the underlying LSM organization. Since LLM workloads fluctuate between cache-population phases and cache-serving phases, the controller adjusts compaction strategies and size ratios dynamically, ensuring that the system sustains high throughput and low latency across workload shifts. This adaptation is performed incrementally to align with natural compaction, avoiding disruptive restructuring.

Finally, Runtime Services provide the operational glue that integrates SGLANG-LSM into LLM serving pipelines. These services include batch-oriented codecs for efficient tensor serialization and compression, automatic background merging of tensor files to manage file system pressure, and streamlined service interfaces that compose naturally with prefix-aware schedulers. Together, these services insulate inference frameworks from low-level storage complexity while ensuring robust performance under dynamic multi-tenant workloads.

Through this architectural decomposition, SGLANG-LSM provides a principled system framework: the storage engine preserves prefix semantics, the controller adapts to workload variability, and the runtime services maintain operational efficiency. This separation of concerns allows each component to evolve independently while collectively supporting scalable, disk-based KV cache management.

## 3.2 PREFIX-PRESERVING STORAGE ENGINE

The Prefix-Preserving Storage Engine serves as the foundational storage component of SGLANG-LSM, positioned between the in-memory radix tree and the underlying disk infrastructure. This component implements a three-layer architecture consisting of a service interface, an LSM-backed index, and a tensor-log store, designed to handle the unique access patterns and scalability requirements of LLM KV cache workloads.

The storage engine employs a key-value separation design where the LSM-backed index manages metadata and ordering information, while the tensor-log Store handles bulk tensor payloads. This architectural separation addresses two critical system requirements: maintaining prefix semantics for efficient cache reuse detection, and avoiding the scalability bottlenecks inherent in file-per-object approaches that plague existing disk-based KV cache systems. The LSM-backed index component stores compact metadata records containing file identifiers and byte offsets. Keys are encoded to preserve lexicographic ordering that corresponds to token prefix relationships, enabling efficient prefix matching operations required by the upper-layer radix tree. The tensor-Log store component manages immutable KV tensor payloads through sequential append-only files, providing the storage substrate for the actual cached computations.

The service interface layer exposes three primary operations to the system: `put_batch` for storing multiple sequential KV caches, `probe` for prefix matching queries, and `get_batch` for efficient retrieval of related cache entries. These operations are designed to align with the batch-oriented access patterns of LLM inference while supporting the prefix-aware semantics required by the radix tree component. Write operations follow a two-phase protocol where tensor data is first committed to the tensor-log store, followed by atomic metadata insertion into the LSM-backed Index. This ordering ensures consistency during system failures and supports the transactional semantics expected by the inference scheduler. Read operations leverage the LSM-tree's range scan capabilities to efficiently locate adjacent prefixes and batch tensor retrievals from the log store. SGLANG-LSM extends SGLang by implementing these operations as a drop-in replacement for SGLang's existing disk storage interface, preserving SGLang's RadixAttention mechanism and in-memory radix tree while seamlessly integrating the LSM-based storage backend without requiring modifications to the upper-layer inference logic.

Unlike file-per-object layouts that suffer from file system metadata overhead and poor spatial locality, the storage engine's design bounds the number of files while preserving prefix relationships. The LSM-backed index handles millions of cache entries through logarithmic lookup complexity, while the tensor-log Store provides predictable I/O patterns through sequential access. Compaction operations in the index layer never touch tensor payloads, ensuring write amplification remains bounded regardless of cache size. This component architecture enables SGLANG-LSM to scale beyond the

limitations of existing approaches while maintaining the semantic structure required for effective prefix-based cache reuse, forming the foundation upon which the adaptive controller and runtime services can optimize system behavior.

## 3.3 ADAPTIVE CONTROLLER

The Adaptive Controller functions as the intelligence layer of SGLANG-LSM, operating above the Prefix-Preserving Storage Engine to continuously optimize system performance under varying workload conditions. This component addresses a fundamental challenge in LLM serving systems: the dynamic nature of cache access patterns that fluctuate substantially over time, shifting between cache-intensive serving phases where lookups dominate, and cache population phases characterized by heavy write workloads.

As observed in recent studies Wang et al. (2025a;b); Zhang et al. (2025), the degree of cache reuse fluctuates substantially over time, shifting between cache-intensive serving phases where lookups dominate, and cache population phases characterized by heavy write workloads. Simultaneously, LSM-trees' performance characteristics are highly sensitive to workload composition changes Dayan & Idreos (2018); Liu et al. (2024a); Mo et al. (2023); Yu et al. (2024a), where different read/write ratios favor different structural configurations. To address these challenges and minimize I/O overhead across varying workload phases, SGLANG-LSM implements workload-aware dynamic compaction that adaptively adjusts LSM-tree parameters in response to observed access patterns.

The controller operates by continuously monitoring workload characteristics and optimizing LSM-tree structural parameters to minimize the weighted I/O cost of all operations. The system models overall I/O cost using theoretical analysis of LSM-trees, specifically tailored for KV cache workloads: $w \cdot W + s \cdot S + r \cdot R + z \cdot Z$, where the cost coefficients $w$, $s$, $r$, and $z$ correspond to the proportions of different operations observed in the current workload window. The individual operation costs $W$, $S$, $R$, and $Z$ are derived from the cost breakdown presented in Section 2. The workload coefficients are extracted directly from SGLANG-LSM's operational statistics by monitoring system behavior. Specifically, $w$ represents the proportion of write operations where new KV cache entries are stored to the LSM-tree, $q$ and $r$ are both derived from successful read operations that retrieve existing cached entries from storage, and $z$ corresponds to probe operations that check for cache entries but find no matching data exists in the system.

The workload coefficients are extracted directly from SGLANG-LSM's operational statistics by monitoring system behavior. Specifically, $w$ represents the proportion of write operations where new KV cache entries are stored to the LSM-tree, $q$ and $r$ are both derived from successful read operations that retrieve existing cached entries from storage, and $v$ corresponds to probe operations that check for cache entries but find no matching data exists in the system.

To find the optimal parameter configuration, SGLANG-LSM iterating over different values of the size ratio $T$ and the runs parameter $K$. Once optimal parameters are determined, SGLANG-LSM employs a lazy transition strategy to minimize the overhead of parameter changes. Rather than immediately restructuring the entire LSM-tree, we incrementally adjust the size ratio during natural compaction operations and modify the number of allowed runs per level as new data is merged. This gradual adaptation approach ensures that structural changes align with the LSM-tree's natural evolution, avoiding expensive reconstruction operations while still adapting to workload shifts.

The workload monitoring operates over sliding windows to capture recent access patterns without accumulating stale statistics. SGLANG-LSM maintains separate counters for $W$, $Q$, $R$, and $V$ within configurable time windows, typically spanning thousands of operations. When significant changes in the workload distribution are detected (using threshold-based detection similar to CAMAL's approach), the system triggers parameter re-optimization using the current window's statistics. This windowed approach prevents the system from over-reacting to temporary fluctuations while remaining responsive to genuine workload transitions.

Through this workload-aware approach, SGLANG-LSM can dynamically balance between configurations optimized for write-heavy cache population phases and read-heavy cache serving phases. During periods of high cache miss rates and frequent insertions, the system favors tiering-like configurations with higher $K$ values to reduce write amplification. Conversely, when cache hit rates are high and lookups dominate, the system shifts toward leveling-like configurations with lower $K$ values to optimize read performance. This adaptive behavior reduces overall I/O costs while maintaining the transition overhead at manageable levels through the lazy adaptation strategy.

## 3.4 RUNTIME SERVICES

The runtime services component operates as the operational layer of SGLANG-LSM, providing essential system-level optimizations and integration capabilities that bridge the gap between the core storage architecture and production deployment requirements. SGLANG-LSM builds upon SGLang Zheng et al. (2024) for LLM inference and RocksDB Facebook (2016) for LSM-tree storage to ensure production readiness. We preserve SGLang's optimized RadixAttention mechanism and in-memory radix tree structure while replacing its disk storage backend with our LSM-based solution. Under continuously growing scale where cache repositories expand indefinitely, the core storage engine alone would still encounter critical bottlenecks including storage explosion and file system limitations that could render the system inoperative. To address these further scalability challenges, we provide the following runtime services:

**Batch Codec Operations.** Recent studies show that compressing KV cache tensors can reduce storage by 50-75% with minimal accuracy impact Sheng et al. (2023); Raistrick et al. (2024). However, traditional file-per-object approaches struggle with efficient batch compression since token-level storage requires additional memory copying and CPU overhead that can negate compression benefits. SGLANG-LSM naturally supports compression through its batch-oriented design. During batch put operations, we compress entire combined tensors before storage, while batch get operations load and decompress tensor blocks together, eliminating the overhead challenges of token-level approaches.

**Automatic Tensor File Merging.** While SGLANG-LSM significantly reduces file counts compared to file-per-object approaches, unbounded growth can still create file system bottlenecks through metadata overhead and reduced I/O efficiency. SGLANG-LSM implements background automatic tensor file merging that activates when file counts exceed configured thresholds. When triggered, merging operations occur during scheduled compaction cycles to avoid interfering with request processing. The system combines adjacent tensor files into consolidated files and updates the corresponding $file\_id + offset$ values in the LSM-tree. This approach leverages existing I/O resources while maintaining data organization, ensuring sustained performance under continuously growing cache workloads.

## 4 EVALUATION

### 4.1 EXPERIMENTAL SETUP

**Hardware.** We evaluate SGLANG-LSM on a server equipped with an Intel Xeon 4314 processor (16 cores at 2.4 GHz), 64 GB of host memory, an NVIDIA A30 GPU (24 GB), an 8 TB SSD, and PCIe 4.0 interface (64 GB/s bandwidth). This configuration represents a typical local deployment scenario LLM serving systems.

**Models.** We evaluate on three representative models: GLM-4-8B and GLM-4-32B GLM et al. (2024), and Llama-3-8B Touvron et al. (2023). These models cover a reasonable size range for local platform deployment and exhibit different KV cache sizes: 40KB, 60KB, and 120KB respectively, allowing us to assess scalability across varying cache object sizes

**Workload.** We employ a synthetic workload that manipulates the expected hit rate to simulate dynamic cache access patterns commonly observed in production environments. The workload progresses through 10 stages with expected hit rates of [0.2, 0.3, 0.5, 0.7, 0.5, 0.3, 0.1, 0.3, 0.5, 0.7], where each stage processes 1000 requests. The expected hit rate is calculated as the ratio of shared prompt tokens to total prompt tokens. We evaluate three different prompt lengths: 4k, 8k, and 16k tokens to assess performance across varying sequence lengths. Before the actual test workload, we run a warmup phase containing 100 million tokens' worth of KV cache, using SGLang's write-through mode to populate both the file backend and SGLANG-LSM disk storage.

**Baselines.** We compare SGLANG-LSM against two SGLang configurations: (1) SGLang with in-memory KV cache management (SGLang(memory)), representing the memory-constrained baseline, and (2) SGLang with file backend (SGLang(file)), representing the current state-of-the-art disk-based approach. We adopt SGLang's default memory setting where GPU memory allocated for KV cache management equals the total GPU memory minus model size and computation overhead, while CPU memory for KV cache is set to twice the GPU allocation.

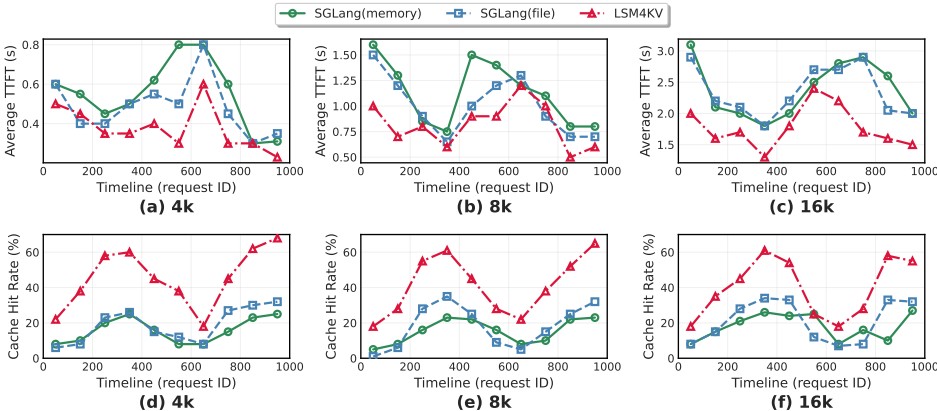

Figure 4: Overall performance of SGLANG-LSM with different prompt length.

**Metric.** We report the average Time-to-First-Token (TTFT) and actual cache hit rate for each stage of the workload. TTFT measures the latency between request submission and first token generation, while cache hit rate indicates the effectiveness of KV cache reuse.

## 4.2 OVERALL PERFORMANCE

**Cache Hit Rate.** As shown in As shown in Figure 4, SGLANG-LSM demonstrates substantial improvements in cache hit rate across all configurations, compared to SGLang with file backend and memory-only storage. Specifically, SGLANG-LSM achieves a cache hit rate that is more than 26 percentage points higher than the file backend approach with 4k prompt length (45.4% vs. 18.7%). The superior performance stems from SGLANG-LSM's ability to leverage LSM-tree's ordered storage capabilities to accommodate significantly more KV cache data, dramatically improving scalability compared to file-per-object approaches that suffer from file system bottlenecks.

The file backend approach encounters severe scalability limitations at large scale, with our experimental platform experiencing file system write anomalies and degraded read performance at about 7 million files. This bottleneck severely constrains the number of KV cache entries that can be effectively stored and retrieved, resulting in substantially lower cache hit rates. Meanwhile, the memory-only approach demonstrates fundamental capacity limitations, as SGLang's default memory allocation can only maintain a small fraction of the total cached data, leading to frequent evictions. In large-scale scenarios where tokens become increasingly dispersed, SGLANG-LSM's ability to store substantially more cache entries than both alternatives translates to consistently higher hit rates.

**TTFT.** SGLANG-LSM consistently achieves lower average TTFT across all prompt lengths, representing significant improvements over SGLang with file backend and memory-only storage. The most substantial improvement occurs with 16k prompts, where SGLANG-LSM reduces TTFT by 24.3% compared to the file backend approach (1.78s vs. 2.35s). The primary driver is SGLANG-LSM's substantially higher cache hit rate, as recomputation time significantly exceeds cache reuse time in our experimental platform, making higher hit rates directly translate to lower overall TTFT.

The benefits of SGLANG-LSM become more pronounced with longer prompt lengths, with absolute TTFT reductions growing from 0.10s (4k) to 0.24s (8k) to 0.57s (16k) compared to the file backend. This scaling effect occurs because SGLang must perform segmented prefill operations for longer prompts under GPU memory constraints, requiring more scheduling operations and memory management overhead. SGLANG-LSM's improved cache hit rate helps reduce these overheads by avoiding redundant computation more frequently, demonstrating that the benefits compound with the computational complexity of longer sequences.

## 4.3 CASE STUDY

**Different LLMs.** As demonstrated in Figure 5(a) and (b), SGLANG-LSM consistently achieves superior performance across different model architectures, with TTFT reductions ranging from 13% to 24% compared to SGLang with file backend across various prompt lengths. Specifically, for GLM-32B with 16k prompts, SGLANG-LSM achieves 8.0s TTFT compared to SGLang(file)'s 9.6s, representing a 16.7% improvement. However, the relative performance gains are more modest for

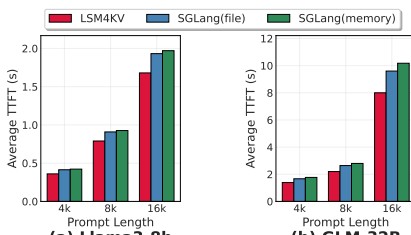 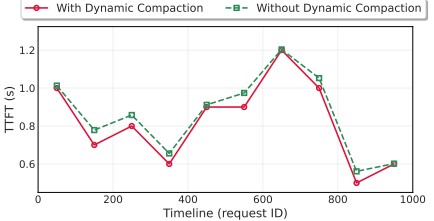

Figure 5: (a)(b) Different LLMs case study. (c) Dynamic compaction case study

Llama3-8B, where improvements range from 13% (4k prompts: 0.36s vs. 0.41s) to 15% (16k prompts: 1.68s vs. 1.93s). This reduced improvement stems from Llama3's substantially larger KV cache tensors (120KB per token compared to 40KB and 60KB for GLM models), while the computational cost of recomputation remains relatively unchanged. As the size disparity between cached tensors and recomputation overhead decreases, the cost differential between cache reuse and recomputation diminishes, thereby reducing the magnitude of TTFT improvements achievable through higher cache hit rates. This demonstrates that SGLANG-LSM's benefits are most pronounced in scenarios where the cost advantage of cache reuse over recomputation is substantial.

**Dynamic Compaction.** Figure 5(c) illustrates the impact of our workload-aware dynamic compaction mechanism on GLM-9B with 8k prompts. SGLANG-LSM with dynamic compaction achieves an average TTFT of 0.81s compared to 0.85s without dynamic compaction, representing a 5% latency reduction. More significantly, the benefits of dynamic compaction are most pronounced during periods of lower baseline TTFT (request IDs 150-400 and 850-950), when cache hit rates are typically lower and workloads exhibit write-heavy characteristics. During these phases, dynamic compaction reduces TTFT by up to 14%, demonstrating that our adaptive approach provides greater optimization for cache population phases compared to read-intensive serving phases. Currently, KV cache write throughput is constrained by the underlying LLM system's inference latency, which limits the full potential of write optimization techniques. As LLM inference systems continue to evolve and achieve higher throughput, we anticipate that dynamic compaction and SGLANG-LSM's overall architectural advantages will become increasingly impactful for large-scale KV cache management scenarios.

## 5 RELATED WORK

**KV Cache Management.** Many LLM inference systems implement KV cache offloading from GPU to CPU and disk storage Kwon et al. (2023); Zheng et al. (2024); Sheng et al. (2023); Liu et al. (2024b); Ye et al. (2024); Qin et al. (2024); hf3fs.. However, these systems typically do not address the scalability challenges of large-scale KV cache storage on local disk. Several works employ selective policies Yao et al. (2024); Gao et al. (2024); Chen et al. (2025); Gim et al. (2024); Raistrick et al. (2024); Agarwal et al. (2025) to reduce KV cache loading and minimize I/O latency. These approaches are orthogonal to SGLANG-LSM and can be combined with our system to enable further KV cache reuse optimization.

**LSM-Tree Store.** Extensive research has focused on LSM-tree optimization, typically employing theoretical analysis to tune parameters such as size ratio, compaction policy, and Bloom filters Dayan & Idreos (2018); Dayan et al. (2017); Huynh et al. (2021); Liu et al. (2024a); Mo et al. (2023); Luo et al. (2020); Dayan et al. (2022); Dayan, Niv and Idreos, Stratos (2019); Huynh et al. (2023); Yu et al. (2024a;b). Key-value separation techniques have been explored to optimize storage efficiency Chan et al. (2018); Lu et al. (2017); Dai et al. (2020). Dynamic tuning approaches for LSM-trees have also been investigated Yu et al. (2024a); Mo et al. (2023). However, none of these works have applied LSM-tree structures to KV cache management in LLM serving systems. SGLANG-LSM is the first work to adapt LSM-tree storage techniques for large-scale KV cache management.

## 6 CONCLUSION

We present SGLANG-LSM, a database-inspired solution that leverages LSM-tree data structures for scalable KV cache management in LLM serving. SGLANG-LSM significantly improves cache hit rates and reduces time-to-first-token latency through key-value separation design and workload-aware dynamic compaction. Our approach represents the first systematic application of database storage techniques to large-scale LLM cache management, demonstrating substantial performance improvements over existing file-based approaches.

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

```
db = LSM4KV()
# The first request
req_0 = "Who wrote Odyssey?"
token_0 = tokenizer(req_0)
# token_0 = [1, 11644, 5456, 6715, 952, 7759, 29973]
# Generate KV cache during prefill
kvcache_0 = llm.intermediate(token_0)
# Len(kvcache_0) = Len(token_0)
with torch.cuda.stream(put_stream)
    db.put_batch(token_0, kvcache_0)
put_count += len(kvcache_0)

# The second request
req_1 = "Who wrote Hamlet?"
token_1 = tokenizer(req_1)
# token_1 = [1, 11644, 5456, 7904, 1026, 29973]
reuse_token = db.probe(token_1)
probe_empty_count += 1 - len(reuse_kvcache)
# reuse_token = [1, 11644, 5456]
with torch.cuda.stream(get_stream)
    reuse_kvcache = db.get_batch(reuse_token)
get_count += len(reuse_kvcache)
# Recompute KV cache of uncached tokens
recomp_token = token_1 - reuse_token
kvcache_1 = llm.intermediate(recomp_token)
with torch.cuda.stream(put_stream)
    db.put_batch(token_1, kvcache_1)

# Background thread
db.compaction(get_count, put_count)
db.merge_file(get_count, put_count)
```

Basic Operation of LSM4KV (Sec 3.2)

Workload-aware dynamic compaction (Sec 3.3)

Auto file merge (Sec 4)

Figure 6: Interface demo of SGLANG-LSM. Primitives provided by SGLANG-LSM are shown in red.

## A    REPLICABILITY

We modify the storage interface based on SGLang https://anonymous.4open.science/r/SGL-KVS-BE8B/ and provide a Python binding for the LSM-tree engine https://anonymous.4open.science/r/python-rocksdb-4454/.

## B    BASIC OPERATIONS

SGLANG-LSM provides three core operations designed for KV cache workloads, as demonstrated in Figure 6.

**Put Batch** operations handle the storage of multiple sequential KV caches simultaneously. As shown in the first request example, when storing KV caches for consecutive token sequences like `token_0` and `token_1`, the system batch-inserts these sequences as keys along with their corresponding $file\_id + offset$ values into the LSM-tree using write batch operations. Concurrently, the associated KV cache tensors (`kvcache_0` and `kvcache_1`) are serialized into tensor files using contiguous storage layouts optimized for sequential access.

**Probe** operations enable efficient prefix matching for cache reuse detection, as illustrated in the second request where the system needs to determine cache availability for `reuse_token`. This operation uses binary search with LSM-tree point lookups to identify the longest cached prefix that can be reused for incoming requests. The LSM-tree's Bloom filters Bloom (1970) efficiently filter non-existent keys during probe operations, minimizing unnecessary I/O when checking for cache misses.

**Get Batch** operations retrieve multiple related KV caches efficiently, as demonstrated when the system needs to load cached tensors for `recomp_token` sequences. The operation first uses LSM-

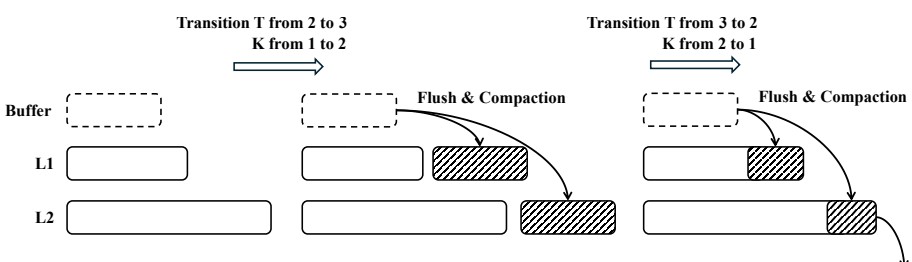

Figure 7: Dynamic compaction in SGLANG-LSM.

tree range scans to retrieve sequences of token identifiers and their corresponding file locations, then performs batch tensor loading from the referenced tensor files. This approach converts what would otherwise be random I/O patterns into efficient sequential reads, significantly improving throughput on disk-based storage systems.

## C DYNAMIC COMPACTION

As illustrated in Figure 7, SGLANG-LSM implements dynamic compaction through lazy parameter transitions that align with the LSM-tree's natural flush and compaction cycles. The system adapts two key parameters: the size ratio (controlling level capacity growth) and the runs parameter (controlling the maximum number of sorted runs per level).

### C.1 PARAMETER TRANSITION MECHANISMS

**Write-Heavy Phase Transition.** When the workload shifts to write-intensive patterns, the system increases both the size ratio and runs parameter to reduce write amplification. During this transition, a key optimization occurs when adjusting the runs parameter: instead of immediately merging all runs at each level, the system allows existing sorted runs from upper levels to be directly moved to lower levels without expensive merge operations. For instance, when the runs parameter changes from one to two at level one, the existing sorted run can remain separate rather than being immediately compacted, significantly reducing sorting overhead during high-write periods.

**Read-Heavy Phase Transition.** When the workload becomes read-dominant, the system reduces both parameters to optimize lookup performance. This transition triggers more aggressive compaction where multiple runs are merged into single sorted runs. When the runs parameter decreases from two to one, the system consolidates separate runs within each level during the next natural compaction cycle, improving read efficiency by reducing the number of runs that must be searched during lookups.

### C.2 IMPLEMENTATION STRATEGY

The lazy transition strategy avoids disruptive full-tree reorganization by leveraging natural compaction opportunities. Size ratio changes are applied when data flows between levels during regular flush operations, while runs parameter changes are implemented during level compaction by adjusting merge policies rather than forcing immediate restructuring. Parameters change gradually over multiple compaction cycles rather than all at once. This approach ensures that structural changes occur as part of the LSM-tree's normal operation, minimizing additional I/O overhead while adapting to workload characteristics. The system can transition between write-optimized configurations for cache population phases and read-optimized configurations for cache serving phases without interrupting ongoing operations.

## D LLM USAGE

LLMs were used exclusively for grammar checking and language editing. All technical content, research ideas, and experimental work are original contributions by the authors.

