# OpenReview forum: "On 10X Better Scalability: KV Stores Scale Up KV Cache"
_ICLR.cc/2026/Conference — Submitted to ICLR 2026_

### Official Review · Reviewer_NMpV · 2025-10-21

**Soundness:** 2
**Presentation:** 2
**Contribution:** 2
**Rating:** 2
**Confidence:** 5

**Summary:**

SGLANG-LSM presents a database-inspired system for scalable Key-Value (KV) cache management in Large Language Model (LLM) serving, adapting Log-Structured Merge-tree (LSM-tree) architectures to address scalability challenges in disk-based cache systems.

**Strengths:**

S1. The key-value separation architecture provides a practical solution to storage scalability.

S2. The system implements a nuanced workload-aware dynamic compaction mechanism, that allow incremental parameter adjustments during natural LSM-tree compactions.

S3. The proposed architecture offers streamlined runtime services that seamlessly integrate with existing LLM serving pipelines

**Weaknesses:**

W1. **Fundamental Performance Limitations**
   Even though there are some solid works on improving the LSM-Trees read performance (e.g., using Learned Index), can the Log-Structured Merge-tree (LSM-tree) fundamentally overcome its inherent read performance challenges? Despite sophisticated optimizations, the multi-level design of LSM-trees introduces significant latency overhead during read operations, with Bloom filters and compaction strategies offering only marginal improvements to the core architectural bottleneck.

W2. **Scalability Concerns**
   How sustainable is the dynamic compaction mechanism under real-world complexity? The intricate adaptive controller introduces substantial system complexity, potentially creating unpredictable performance variations. The continuous workload monitoring and parameter adjustment might generate more computational overhead than the performance gains it promises.

W3. **Experimental Validation Deficiencies**
   The experimental evaluation relies heavily on controlled, synthetic scenarios that fail to capture the nuanced, unpredictable characteristics of real-world LLM serving environments. The narrow experimental scope significantly limits the comprehensive understanding of the system's performance across diverse operational contexts.

W4. **Architectural Complexity Trade-offs**
   Can the key-value separation design justify its computational complexity? While innovative, the architectural approach introduces substantial runtime overhead. As KV cache tensor sizes increase, the performance gains from this design appear to diminish, revealing potential scalability constraints in the system's fundamental approach.

W5. **Methodological Gaps**
   The paper demonstrates an incomplete exploration of LSM-tree characteristics (SST size, block sizes, level multiplier, Bloom filter false-positive rate, using or not using Blob), lacking a robust cost model that can accurately predict system behavior across different workload scenarios. The methodological sound end-to-end foundation need to design to fully validate the proposed approach. A thorough system setup that accounts for all parameters from both the LLM and LSM-Tree perspectives.

W6. **Reproducibility and Code vagueness**
    The  submitted RocksDB code's ReadMe just provides the compilation commands and no more information provided in it, For example, I could not verify the presence of the "dynamic compaction approach"  in the db/compaction directory, specifically within compaction_job which is the ideal location for its implementation.

**Questions:**

Addressing W1 to W6

---

### Official Review · Reviewer_izj8 · 2025-10-30

**Soundness:** 2
**Presentation:** 2
**Contribution:** 2
**Rating:** 2
**Confidence:** 3

**Summary:**

The paper proposes SGLANG-LSM, a disk-resident KV-cache backend for large language model (LLM) serving that replaces the traditional file-per-object layout with a Log-Structured Merge (LSM) tree index and an append-only tensor log. It further introduces an adaptive controller that dynamically tunes LSM parameters (e.g., tiering/leveling, size ratio) based on observed read/write ratios, and implements runtime services for batch tensor codecs and background file merging. Experiments conducted on a single machine (NVIDIA A30 GPU, 64 GB RAM, 8 TB SSD) report a 143% increase in cache-hit rate and up to 24% reduction in time-to-first-token (TTFT) compared to SGLang’s existing memory- or file-based backends, under synthetic workloads.

**Strengths:**

1. The paper correctly identifies a practical scalability problem: existing file-per-object KV-cache backends create millions of small files, leading to metadata overhead and poor I/O locality.
2. The application of LSM-tree storage to KV-cache management is relatively new.
3. The reported 24% TTFT reduction represents a measurable and relevant performance improvement for latency-sensitive LLM inference.

**Weaknesses:**

1. Questionable necessity of LSM for this workload.
Modern KV-cache management systems (e.g., LMCache (Cheng et al., 2025)) already compute hash-chained prefix identifiers and perform O(1) lookups over content-addressed append logs. Since most KV-cache objects are immutable and written once, the motivation for adopting a full LSM-tree (with compaction and sorted order maintenance) is not well justified.
2. Incomplete evaluation details.
The evaluation setup raises questions: the authors report handling 7 million files even though the hardware configuration (A30 + 64 GB RAM) is relatively small. The paper does not specify the KV-cache page size, or the prefix-sharing ratio, all of which are necessary to assess scalability and reproducibility.
3. Misleading title and framing.
The title “10× Better Scalability” is not supported by the presented results.
4. Weak and outdated baselines.
The baseline SGLang (file)—is too naive to substantiate strong claims of improvement. The paper does not compare against state-of-the-art tiered KV-cache systems such as LMCache+vLLM.

**Questions:**

1. Performance scaling beyond 16k tokens.
The TTFT trends for 4k and 8k sequences are counterintuitive: even when cache hit rates are significantly higher than baselines (e.g., 8k at request ID ≈ 250), the TTFT reduction is marginal. Prior work that overlaps KV-cache computation and disk loading (Jin et al., 2024) shows that early tokens' KVCache recomputation is faster than loading, which might be the reason why SGLang-LSM didn't excel in those cases. It will make more sense for SGLang-LSM to evaluate on a longer context (e.g., 32k or 64k)?
2. Scalability across nodes and tiers.
The current experiments are single-node and local-disk only. How would the system extend to remote disks, multi-node environments, or tiered storage hierarchies (e.g., GPU → CPU → SSD → network store)? Such results are critical to substantiate the “scalability” claim.
3. Correlation between LSM and hit rate.
The motivation for using LSM is unclear. If we want to deal with too many small files and increase the capacity, we can concatenate the KVCache and store an index file separately; this would be much easier. I wonder why LSM can increase the hit rate.

---

### Official Review · Reviewer_5yY9 · 2025-11-01

**Soundness:** 2
**Presentation:** 1
**Contribution:** 2
**Rating:** 2
**Confidence:** 3

**Summary:**

In this paper, the authors introduce SGLANG-LSM, a database-inspired system that applies Log-Structured Merge-tree (LSM-tree) storage architectures to manage large language model (LLM) Key-Value (KV) caches) at scale.
Traditional disk-based KV cache systems (e.g., in SGLang),  each token’s cache tensor is stored as a separate file. This approach leads to scalability issues due to metadata overhead, I/O inefficiency, and poor spatial locality.
To address this, the authors design a prefix-preserving LSM storage engine that maintains token sequence locality while using key-value separation to efficiently store large tensors (start tensor files as a log).
An adaptive controller dynamically tunes LSM parameters such as compaction and size ratios based on changing workload patterns, balancing performance between read- and write-heavy phases. Complementary runtime services further enhance scalability through batch operations, tensor compression, and automatic file merging. Experiments across GLM-4-8B, GLM-4-32B, and Llama-3-8B models show up to 143% higher cache hit rates and 24% lower time-to-first-token latency.

**Strengths:**

S1. The paper presents an interesting integration of LSM-tree storage architectures (e.g., RocksDB) into LLM serving (e.g., SGLang). This cross-domain design bridges disk storage and AI systems for improved scalability.

S2. The experiments results seems to be good;

**Weaknesses:**

This paper raises a lot of questions about the design choices and evaluation scope of SGLANG-LSM despite its strong technical contributions.

W1. Since the LSM-tree design is primarily optimized for high write throughput, it is unclear why it was chosen over read-optimized alternatives such as B-trees, especially when the main goal of LLM serving is to accelerate cache retrieval. The authors should better motivate why enabling high write throughput is critical for this workload.

W2. The paper proposes a tensor-log append-only storage layer with periodic merging, yet this design might reintroduce the same spatial locality issues that the authors attribute to file-per-object layouts.

W3. It is unclear how SGLANG-LSM achieves substantially higher cache hit rates than existing approaches when the cache eviction policy remains unchanged. The authors should explicitly define the notion of “cache hit” and explain whether structural improvements alone can increase it without modifying replacement strategies.

W4. The evaluation primarily compares against SGLang HiCacheFile, which the SGLang authors describe it as a simple file-based backend. To strengthen claims of scalability and performance, the system should be compared with stronger baselines such as AIbrix KVCache or LMCache, which are more representative of current production systems.

**Questions:**

Please see weakness.

---

### Official Review · Reviewer_yc3X · 2025-11-02

**Soundness:** 3
**Presentation:** 3
**Contribution:** 3
**Rating:** 6
**Confidence:** 2

**Summary:**

This paper proposes SGLANG‑LSM, a drop‑in disk backend for SGLang that replaces file‑per‑object KV‑cache storage with an LSM‑tree–backed, prefix‑preserving storage engine and a tensor‑log for large payloads, plus an adaptive controller that tunes LSM parameters to the current read/write mix and runtime services for batch codecs and automatic file merging. The design preserves prefix semantics (for longest‑prefix reuse), separates keys and values so compaction never rewrites large tensors, and exposes three operations—put_batch, probe, get_batch. Experiments on a single A30 server with 8 TB SSD show higher cache‑hit rates (e.g., +26.7pp at 4k prompts; 45.4% vs. 18.7%) and lower TTFT (up to 24.3% at 16k prompts: 1.78 s vs. 2.35 s) relative to SGLang’s disk backend; dynamic compaction yields ~5% average TTFT reduction and up to 14% in write‑heavy phases. Figures 1–3 (pp. 2–4) illustrate the architecture, Figure 4 (p. 8) presents TTFT and hit‑rate trends over staged workloads, and Figure 5 (p. 9) shows model‑wise results and the impact of dynamic compaction.

**Strengths:**

* The paper is the first to apply database LSM‑tree designs to LLM KV‑cache management, including a prefix‑preserving key encoding and explicit key–value separation for tensor payloads.
* The paper is well-written and has a clear layered architecture of the system.
* The paper provides a concrete pseudo‑API (Appendix B, Figure 6) that shows how probe and get_batch compose with SGLang’s RadixAttention, which helps reproducibility.

**Weaknesses:**

* The workloads are synthetic with ten staged hit‑rate phases. Results would be more convincing with traces reflecting real-world multi‑turn chat or retrieval‑augmented traffic.
* Beyond SGLang(memory) and SGLang(file), there are no comparisons to other disk‑backed KV‑cache systems (e.g., recent multi‑tier prefix stores) nor ablations teasing apart contributions of prefix‑encoding, key–value separation, dynamic compaction, and runtime services.

**Questions:**

* Please provide ablations isolating (i) prefix‑preserving key encoding, (ii) key–value separation (tensor‑log vs. in‑place values), (iii) dynamic compaction on/off, and (iv) runtime services (compression, auto‑merge) to show each component’s contribution to TTFT and hit rate.
* Beyond SGLang(file), can you compare against at least one other disk‑backed KV‑cache store (e.g., a vanilla RocksDB with flattened keys or a recent multi‑tier prefix store) to substantiate the novelty/benefit claims? Even a careful “strongest possible” file‑backend with batched I/O and compression would help.

---

### Meta-Review · Area_Chair_HR1k · 2026-01-12

**Summary:**

This paper proposes SGLANG-LSM, an LSM-tree–backed disk KV-cache store for SGLang that replaces the current file-per-object backend, aiming to improve scalability and TTFT via a prefix-preserving index + tensor-log separation and an adaptive compaction controller.

However, reviewers raised several strong concerns that remain decisive:
* the “10× scalability” framing appears unsupported;
* the choice of LSM (vs simpler/indexed append-log or read-optimized structures) is not convincingly justified;
* it is unclear why cache hit rate increases if the eviction/replacement policy is unchanged;
* the evaluation uses weak/outdated baselines while the community already has multi-tier KV cache layers (e.g., LMCache, AIBrix KVCache).

**Reviewer Concerns:**

There is no rebuttal, so the main reviewer concerns are still outstanding:

* Over-claiming / misleading framing: the title “10× Better Scalability”.
* LSM motivation is weak: reviewers questioned why an LSM-tree (write-optimized) is the right tool when the serving goal is fast reads, and why alternatives (e.g., B-trees / simpler index+log approaches) are not considered.
* Cache-hit definition / source of gains is unclear:
* Evaluation is narrow + baselines too weak: experiments are single-node, synthetic, and mainly compare against SGLang(file)/HiCacheFile

**Reviewer Scores:**

N/A, since no rebuttal is provided..

---

### Decision · Program_Chairs · 2026-01-26

Reject